# MicroRNA and Protein Biomarkers of Intestinal Permeability in the Assessment of Metabolic Dysfunction-Associated Steatotic Liver Disease (MASLD)

**DOI:** 10.3390/ijms262311351

**Published:** 2025-11-24

**Authors:** Dominika Białek, Ewa Wunsch, Agnieszka Kempińska-Podhorodecka, Joanna Abramczyk, Adam Wunsch, Katarzyna Kozłowska

**Affiliations:** 1Department of Translational Medicine, Pomeranian Medical University, 71-252 Szczecin, Poland; bialekdm@gmail.com (D.B.); ewa.wunsch@pum.edu.pl (E.W.); 2Department of Medical Biology, Pomeranian Medical University, 70-111 Szczecin, Poland; agnieszka.kempinska.podhorodecka@pum.edu.pl (A.K.-P.); joanna.abramczyk@pum.edu.pl (J.A.); adam.wunsch2005@gmail.com (A.W.)

**Keywords:** metabolic dysfunction-associated steatotic liver disease (MASLD), gut barrier dysfunction, microRNA, biomarkers

## Abstract

Intestinal barrier dysfunction and microRNA dysregulation are proposed contributors to progression of metabolic dysfunction-associated steatotic liver disease (MASLD). We aimed to assess selected protein and miRNA biomarkers of intestinal permeability in relation to MASLD severity. We included 104 patients with MASLD and 57 healthy controls. Serum lipopolysaccharide-binding protein (LBP), diamine oxidase (DAO), tumor necrosis factor alpha (TNF-α), interleukin 6 (IL-6), and miRNAs (miR-21, miR-29a, miR-122) were measured. Multivariable logistic regression identified independent predictors of steatosis and fibrosis severity. Patients with MASLD showed higher LBP levels (*p* = 0.002) and increased serum miR-122 expression (*p* < 0.0001) compared with controls. LBP correlated with CAP values (Rho = 0.23, *p* = 0.02) and was elevated in advanced steatosis (*p* = 0.04). DAO levels correlated with CAP (Rho = 0.22, *p* = 0.02) and were higher in advanced steatosis (*p* = 0.04) but decreased in advanced fibrosis (*p* = 0.04). MiR-122 correlated with fibrosis indices (TE: Rho = 0.22, *p* = 0.03; APRI: Rho = 0.41, *p* = 0.0001) and liver enzymes (ALT: Rho = 0.40, AST: Rho = 0.50, both *p* < 0.0001). Logistic regression identified elevated miR-122 and reduced miR-21 as independent predictors of MASLD, while DAO and transaminases predicted advanced steatosis. Elevated serum miR-122, alongside reduced miR-21, independently predict MASLD. DAO is associated with steatosis severity, while miR-122 reflects fibrotic progression.

## 1. Introduction

Metabolic dysfunction-associated steatotic liver disease (MASLD) is nowadays one of the most common chronic liver diseases, affecting up to 30% people globally and increasing, parallel to the growing prevalence of obesity and obesity-related diseases [1].

Since the first introduction of non-alcoholic fatty liver disease (NAFLD), the nomenclature of the disease has undergone many revisions. Because of observed relationship between obesity, lipid metabolism malfunction, insulin resistance, diabetes, and hepatic steatosis and its progression, in 2023, the new term—MASLD—has been introduced [2].

MASLD involves a wide spectrum of different pathological conditions. In most of the patients the course of liver disease is mild with limited risk of progressive liver injury and adverse outcomes. However, in a proportion of affected subjects the disease progresses from simple steatosis to inflammation and fibrosis and eventually end-stage liver disease. The identification of this subpopulation of patients is crucial, as they require most careful monitoring and effective treatment. Liver biopsy remains the golden standard for the diagnosis of hepatic inflammation (metabolic associated steatohepatitis, MASH). However, due to its invasiveness and high cost, the procedure should be optimally performed in patients at risk of the progressive phenotype of the disease. Therefore, noninvasive biomarkers and indexes for the liver injury assessment are developed. Classical markers of hepatocellular injury like transaminases or metabolic show weak correlation with MASLD severity. Therefore, novel markers are actively pursued. Many candidate biomarkers are molecules linked with the pathogenic pathways linked with MASLD progression.

The pathogenesis of liver injury in MASLD involves multiple factors including insulin resistance, dysregulation of adiponectin, oxidative stress, genetic predisposition, and environmental factors. Among the latter, a Western-style high-fat diet has been implicated in promoting not only obesity but also intestinal dysbiosis with subsequent chronic inflammation and impaired intestinal barrier function [3,4], which results in increased permeability to harmful microbiological agents. These substances, upon reaching the liver via the portal circulation, may induce inflammatory responses leading to fibrosis and cirrhosis [5]. The significance of disturbances within the gut–liver axis has been postulated in the pathomechanism of various liver diseases, ranging from those of alcoholic etiology, through autoimmune origins, and finally to MASLD.

In fact, previous studies have shown that serum markers of intestinal permeability are elevated in patients with MASLD, even in early stages of the disease, when compared to healthy population [6]. Nevertheless, data coming mostly from animal models suggest that the disruption of intestinal barrier in liver steatosis is not just a coincidence, but most importantly plays a crucial role in the development of MASLD, promoting the progression from simple steatosis to the progressive forms of the disease [6,7].

Currently, widely accepted studies aimed at assessing intestinal barrier permeability measure the urinary excretion of orally administered marker molecules (Cr-EDTA) or disaccharides (e.g., the lactulose–mannitol test), or evaluate intestinal segments ex vivo by Ussing chambers. These methods, however, require substantial time investment or specialized techniques that are not available in most research centers. Hence, the measurement of serum biomarkers that may reflect the extent of intestinal barrier damage is becoming increasingly widespread. Intestinal fatty acid–binding protein (I-FABP) is expressed in the epithelial cells of the mucosal layer of the small intestine. Its increased concentrations have been observed in intestinal ischemia [8] and celiac disease [9]. However, its levels did not correlate with the results of the lactulose/mannitol ratio [10]. Another biomarker, zonulin, has been widely used as an indicator of intestinal permeability. Elevated levels have been reported in intestinal diseases such as celiac disease. Nevertheless, subsequent studies have advised caution in interpreting serum zonulin measurements as a marker of intestinal barrier integrity due to methodological inaccuracies as the most commonly used enzyme-linked immunosorbent assays (ELISA) were later shown to detect proteins other than zonulin [11]. Lipopolysaccharide (LPS) is a protein found in the outer membrane of Gram-negative bacteria, used most commonly as a marker of bacterial endotoxemia [3].

LBP is widely utilized as a marker of LPS exposure and chronic low-grade inflammation [12]. Its serum concentrations have been shown to correlate with established methods of assessing intestinal permeability (such as lactulose/mannitol ratio), making it a promising marker of gut barrier function [10]. DAO, a secretory protein of intestinal epithelial cells, has been recently postulated as a novel serum marker of intestinal barrier impairment [13].

MicroRNAs are small noncoding RNA molecules that regulate gene expression post-transcriptionally, influencing a wide range of physiological and pathophysiological processes. To date, numerous studies have highlighted the role of miRNAs in the pathogenesis of various diseases, by analyzing their tissue expression and levels of circulating miRNA in blood and serum.

In patients with inflammatory bowel disease or irritable bowel syndrome with diarrhea, several microRNAs—including miR-16, miR-21, miR-29, miR-122a, miR-125b-5p, miR-146b-5p, miR-155, and miR-223—have been postulated to influence tight junctions in the intestinal epithelium by modulating the expression of proteins such as zonulin, occludin, various claudins, and other TJ proteins, thereby contributing to intestinal barrier dysfunction [14].

In this study, we aimed to investigate the correlation between serum markers of intestinal permeability (lipopolysaccharide binding protein (LBP), diamine oxidase (DAO)), as well as selected miRNAs, and the severity of MASLD in a well-characterized group of patients, in order to evaluate their usefulness in the non-invasive assessment of disease severity. Importantly, our study provides a novel integrative approach by combining established protein-based indices of intestinal permeability with circulating miRNAs, which although previously proposed as hepatocyte-specific molecules (miR-122) [15] or potential markers of intestinal barrier dysfunction (miR-21 and miR-29a) [16,17] have not yet been jointly assessed in the context of MASLD severity. For the assessment of systemic inflammation, we also measured (interleukin 6 (IL-6) and tumor necrosis factor (TNF-α)).

## 2. Results

### 2.1. Study Population

Among the 127 patients with MASLD initially screened, 10 were excluded due to invalid controlled attenuation parameter (CAP) measurements or meeting the exclusion criteria. Additional thirteen individuals were disqualified from the study due to missing biochemical test results or insufficient serum volume. A total of 104 patients (42 males, median age 55 years) were included in the final data analysis. Within the study group, two subgroups were identified based on the degree of liver steatosis, according to Karlas et al., as such: the first subgroup included 46 patients with mild to moderate steatosis (S1–S2), while the second subgroup comprised 58 patients with significant steatosis (S3) (Figure 1).

Another classification within the study group was made based on the degree of liver fibrosis assessed by transient elastography. This classification consisted of 87 patients without significant fibrosis (F0–F1) and 15 patients with moderate to advanced fibrosis (F2–F4).

The characteristics of MASLD patients and sex and age-matched controls are summarized in Table 1.

### 2.2. Markers of Intestinal Permeability in Study and Control Group

LBP serum levels were significantly higher in the study group compared to the control group (*p* = 0.002). There was no significant difference in serum concentrations of TNF-α, IL-6, or DAO between the groups. Regarding microRNA, there was a significantly increased expression of serum miR-122 levels in MASLD patients compared to controls (*p* < 0.0001). No significant differences were observed in miR-21 and miR-29a levels between the two groups. These data are presented in Table 2.

### 2.3. Correlation Between Protein Markers of Intestinal Permeability and MASLD Severity

In the study group, serum LBP level was positively associated with CAP values (Rho = 0.23, *p* = 0.02; Table 3) and was significantly higher in advanced steatosis grade estimated by CAP (median 22.9 μg/mL in S1–S2 group vs. median 33.2 μg/mL in S3 group, *p* = 0.04; Table 4). There was no significant association between LBP levels and the presence or stage of liver fibrosis (Table 4). Besides a negative association with bilirubin levels (Rho = 0.25, *p* = 0.01), no other correlation was found between LBP concentrations, liver biochemistry, and metabolic parameters (Table 3).

Circulating DAO concentrations were positively associated with liver steatosis severity assessed with CAP (Rho = 0.22, *p* = 0.02; Table 3) and its degree (38.9 ng/mL in S1–S2 group vs. 49.0 ng/mL in S3 group, *p* = 0.04; Table 4). Nevertheless, DAO levels were negatively associated with liver fibrosis indexes: aspartate aminotransferase (AST) to platelet ratio (APRI) (Rho = −0.26, *p* = 0.02) and Fibrosis-4 (FIB-4) (Rho = −0.37, *p* = 0.0007) (Table 3) and lower DAO concentrations were associated with more advanced fibrosis assessed in transient elastography (44.2 ng/mL in F0–F1 group vs. 28.9 ng/mL in F2–F4 group, *p* = 0.04; Table 4). No significant correlations were found between DAO levels and laboratory metabolic or liver tests (Table 3).

There was no significant association between IL-6 and TNF-a concentration and analyzed parameters of the MASLD severity (Table 3 and Table 4).

### 2.4. Correlation Between Microrna and MASLD Severity

Despite miR-122 showing a positive correlation with non-invasive steatosis indices derived from patients’ anthropometric measurements and laboratory results including fatty liver index (FLI) (Rho = 0.28, *p* = 0.005) and ultrasound (US) assessment as in hepatorenal index (HRI) (Rho = 0.22, *p* = 0.02), no significant association was found with the liver steatosis as assessed by CAP (Rho = 0.17, *p* = 0.08) (Table 3). Nevertheless, a correlation was observed between miR-122 levels and the fibrosis assessed by transient elastography (TE) (Rho = 0.22, *p* = 0.03) and APRI (Rho = 0.41, *p* = 0.0001) (Table 3), as well as a borderline association with fibrosis severity (4.8-fold increase in the F0-F1 group vs. 7.6-fold increase in the F2–F4 group, *p* = 0.05; Table 4). Finally, there was a positive significant association between miR-122 and aminotransferases, alanine aminotransferase (ALT) (Rho = 0.4, *p* < 0.0001) and AST (Rho = 0.5, *p* < 0.0001), gamma-glutamyl transferase (GGT) (Rho = 0.27, *p* < 0.005), bilirubin and ferritin levels (*p* < 0.05) (Table 3).

In the study group, there was no observed association between stages of liver steatosis or fibrosis and levels of miR-21 or miR-29a (Table 4).

Remarkably, levels of fasting insulin were also correlated positively with steatosis degree (18 U/mL In S1–S2 group vs. 16 U/mL in S3–S4 group, *p* = 0.0001) and fibrosis severity (13 in F0–F1 group vs. 27 in F2–F4 group, *p* = 0.0006). Similar associations were observed with aminotransferases (Table 3).

### 2.5. Independent Indicators of the Presence of Liver Steatosis

In order to investigate whether analyzed markers of intestinal permeability may serve as novel serum non-invasive discriminators of liver steatosis in non-diagnosed population, we performed multivariable logistic regression analysis. The data derived from control and study group were included. The final model revealed increased miR-122 (OR = 1.1, 95%CI = 1.01–1.2, *p* = 0.029), GGT (OR = 1.05, 95%CI = 1.02–1.09, *p* = 0.001), and ferritin (OR = 1.008, 95%CI = 1.002–1.01, *p* = 0.005) as well as decreased miR-21 (OR = 0.49, 95%CI = 0.279–0.874, *p* = 0.015) as independent predictors of liver steatosis in MASLD. These data are presented in Figure 2 and in Appendix A.

### 2.6. Independent Predictors of Severe Steatosis

In order to investigate whether analyzed parameters are independent predictors of severe steatosis in MASLD we performed multivariable logistic regression analysis with stepwise backwards selection on the data derived from the study group. In the final model higher DAO concentration (OR=1.03, 95%CI=1.010–1.044, *p* = 0.002), alongside with transaminases activity were independent predictors of advanced steatosis indicated by CAP. These data are presented in Figure 3 and Appendix A.

### 2.7. Independent Predictors of Advanced Fibrosis

In the univariable analysis there was an association between fibrosis severity assessed by TE and ALT, AST, insulin, DAO and miR-122 levels (Table 2). However, in the logistic regression analysis only transaminases activity remained independent predictors of advanced liver fibrosis (Appendix A).

### 2.8. Associations Between Analyzed Parameters of Intestinal Permeability

Circulating DAO concentrations were positively correlated with LBP levels (*p* = 0.02). miR-21; however, they correlated significantly with TNF-α concentration. All three examined circulating miRNA were significantly positively correlated with each other (*p* < 0.001). These data are presented in Appendix A.

## 3. Discussion

In this study, we examined serum markers of intestinal permeability in patients with MASLD to assess their potential utility for non-invasive evaluation of the presence and severity of liver injury. Both established protein markers recognized as indicators of impaired gut barrier integrity (LBP, DAO) and several emerging miRNA biomarkers (miR 21, miR 29a, miR 122) were analyzed in the context of liver biochemistry and the severity of steatosis and fibrosis.

Serum LBP is proposed as a promising marker of increased intestinal permeability and an indirect indicator of endotoxemia [12]. Some studies have demonstrated strong correlation between LBP levels and well-established but time-consuming methods of assessing intestinal barrier permeability, such as the lactulose/mannitol excretion ratio. What is more, this correlation was independent of age, body mass index (BMI), and sex [10]. It has been suggested that LBP plays a role in binding lipopolysaccharide and facilitating its transfer to toll-like receptor 4 (TLR4) complex [19,20], which initiates an extensive cell signaling pathway leading to an inflammatory response and cytokine expression and secretion, and is considered a critical factor in NAFLD development [20,21].

In the present study, LBP was identified as an independent predictor of the presence of liver steatosis, which may support the hypothesis that increased intestinal barrier permeability is one of the contributing factors to the development of liver steatosis. This finding aligns with previous studies that have demonstrated this relationship using different methods for assessing intestinal barrier function, such as the urinary excretion of the 51Cr-EDTA test, immunohistochemical analysis of zona occludens-1 (ZO-1) expression in duodenal biopsy specimens, or measuring serum LPS [21,22]. These studies have also reported no association between intestinal barrier dysfunction and the development of non-alcoholic steatohepatitis (NASH), contrary to other publications describing not only a link between increased intestinal permeability and NASH [7,23,24], but also the absence of such an association with simple steatosis [25]. In contrast, other studies found that intestinal permeability, measured by LPS concentration, correlated with the severity of steatosis, with differences in LPS levels observed not only between stages of simple steatosis, but also between simple steatosis and NASH [24,26]. Furthermore, in a big group of patients’, serum LPS was independently associated with future incidents of advanced liver disease in the general population [27].

In the present study, patients with more severe steatosis had higher LBP serum concentration than patients with steatosis grade 1–2, but LBP was not proven to be an independent factor for severe steatosis. Moreover, in the current analysis, no association was found between LBP and BMI in patients with diagnosed MASLD. This finding differs from previously published data, which reported a positive correlation between LBP levels and body weight in patients with metabolic syndrome [28,29]. Moreover, prior studies have described a decrease in LBP concentration in obese patients following weight loss achieved through a low-calorie diet [28,30] or bariatric surgery [31].

Furthermore, LBP levels did not correlate with fasting glucose or insulin concentrations in the study group. Some previous studies have identified insulin resistance as a factor contributing to increased intestinal permeability in individuals with obesity [32]. Additionally, increased intestinal permeability measured using the lactulose/mannitol ratio, which was linked to elevated homeostatic model assessment (HOMA) index values in obese patients, has been shown to improve following significant and successful weight reduction [33].

These LBP findings may be influenced by other factors affecting its concentration. The exact role of this protein in the pathophysiology of liver steatosis remains a subject of ongoing debate. For example, in mice fed a high-fat, high-sucrose diet, the deletion or knockdown of the *Lbp* gene led to significant improvement of liver steatosis by attenuating diet-induced hepatic lipogenesis, fibrosis, and inflammation-related pathways [34]. However, in the absence of obesity or diet-induced hepatic lipogenesis, liver-derived LBP appeared to have a protective function, helping to prevent liver inflammation, oxidative stress, and fibrosis, which suggests that inhibiting LBP could potentially amplify the proinflammatory effects of LPS [20,35]. In a clinical study, obese patients with NAFLD exhibited lower serum concentrations of LBP compared with obese individuals without hepatic steatosis. Moreover, reduced serum LBP levels were associated with type 2 diabetes mellitus independently of BMI, age, and sex. These findings may be explained by the proposed protective role of LBP, which, in complex with HDL3, prevents LPS from binding to and activating hepatic macrophages, thereby preventing the inflammatory response and ultimately conferring protection against metabolic diseases [36,37]. This theory may partially explain the findings of the present study, in which no correlation was observed between LBP levels and either BMI or disturbances in glucose metabolism among patients with MASLD.

DAO is a secretory protein found in the cytoplasm of intestinal epithelial cells. Damaged cells release DAO into the bloodstream, increasing its normally low and stable serum concentration. Therefore, an increase in DAO concentration has been postulated as a novel marker of intestinal barrier impairment [13].

In present study, serum DAO concentration was independently and positively correlated with the severity of liver steatosis, which aligns with recent findings showing that DAO levels depend on the degree of hepatic steatosis [13]. This may also serve as further evidence of the association between MASLD and impaired intestinal barrier function, as DAO is a protein found in the cytoplasm of intestinal epithelial cells. Elevated plasma DAO levels indicate the repair process of intestinal damage, and increased concentrations have been observed in conditions such as small intestinal obstruction, superior mesenteric artery occlusion, and active Crohn’s disease [38]. The link between DAO and intestinal barrier function appears to be further supported by the observed association between DAO and LBP levels in the present study.

To the best of our knowledge, this study is the first to investigate DAO concentrations in MASLD-associated fibrosis. Interestingly, in patients with advanced fibrosis (F2–F3), DAO levels were lower compared to those with F1–F0 stage. In cirrhosis, existing reports on DAO concentrations suggest an increase in its levels in this condition [39]. Elevated DAO has been correlated with hospital readmission in patients with cirrhosis, indicating a potential association between DAO and the risk of liver disease decompensation with intestinal barrier dysfunction as an important factor [40].

Decreased DAO levels have thus far been reported in acute mesenteric ischemia [38] and have been shown to correlate with reductions in villus length and surface area during anticancer drug treatment [41]. These observations support another concept of DAO as a biomarker reflecting the number of mature, functional enterocytes. However, they do not explain the decrease in DAO levels observed in patients with advanced liver fibrosis. Given the small number of participants with hepatic fibrosis in the current study, further research involving a larger cohort is warranted.

MicroRNA-122 is considered a hepatocyte specific miRNA, which constitutes 70% of the entire miRNA pool in the liver [15]. In multiple studies serum miR-122 was elevated in patients with NAFLD and was proposed as a potential biomarker for distinguishing NAFLD patients from healthy controls [42]. Moreover, miR-122 has been reported to play a role in the regulation of lipid metabolism, with elevated circulating free fatty acid levels—associated with insulin resistance—stimulating its hepatic production [43].

Our study confirms previous reports indicating an elevation of serum miR-122 levels in patients with MASLD compared to those without hepatic steatosis. According to numerous studies, circulating miR-122 is not only elevated in NAFLD patients but also correlates with the severity of liver steatosis on histopathological examination and the degree of inflammation, reaching even higher levels in NASH [42].

Despite the significant difference in miR-122 levels between MASLD and non-MASLD patients in our study, we did not find a correlation between miR-122 levels and hepatic steatosis severity, as assessed by CAP. However, the significant correlation between miR-122 levels and other indices of hepatic steatosis, such as FLI and HRI, may partially reflect this relationship.

Circulating miR-122 levels increased with the progression of liver fibrosis, supporting previous findings that miR-122 is one of the microRNAs associated with both the presence and severity of fibrosis in NAFLD [44]. One study suggests that changes in circulating miR-122 could serve as a useful predictor of prognosis in NAFLD patients with severe fibrosis stage and no improvement in stage scores [45]. Other studies describe a decline in circulating miR-122 levels in patients with cirrhosis, despite its initial elevation in NAFLD and NASH [46].

The role of miR-21 in MASLD has not been clearly established yet. One of the proposed pathophysiological mechanisms of MASLD involves a link between disease development and endotoxemia caused by intestinal barrier dysfunction. MiR-21 has been frequently suggested in the literature as a factor associated with intestinal barrier impairment. Studies using filter-grown Caco-2 monolayers demonstrated a significant elevation in miR-21 expression under conditions of increased tight junction barrier permeability induced by TNF-α. Furthermore, miR-21 overexpression significantly increased intestinal permeability, while miR-21 knockdown had the opposite effect [47,48]. In multiple studies of patients with ulcerative colitis and Crohn’s disease miR-21 found to be upregulated in intestinal tissue [49], in serum [50], and feces [51]. Conversely, other findings indicated that miR-21 may serve as a protective factor against intestinal barrier dysfunction by promoting the expression of tight junction proteins through targeting rho-associated coiled-coil kinase 1 (ROCK1) both in vivo and in vitro [47].

Upregulation of miR-21 in the liver has been observed across various hepatic diseases, including MASLD. MiR-21 operates through a complex transcriptional network to regulate glucose and lipid metabolism in hepatocytes and may contribute to the development of steatosis by several mechanisms [52,53,54]. In the context of fibrosis, a more direct association between miR-21 and the activation of hepatic stellate cells (HSCs) has been documented [55]. MiR-21 upregulation has been reported in both hepatocytes [52] and serum [56] from patients with steatohepatitis as well as those with fatty liver disease associated with metabolic disorders [57]. However, contrasting findings from a separate study indicated that serum miR-21 levels were significantly lower in 25 NAFLD patients compared to 12 healthy controls [58].

The present study revealed that in the MASLD patient group, circulating miR-21 levels were lower compared to healthy individuals, and the decrease in its concentration was one of the independent predictors of MASLD. This result is consistent with the aforementioned study [58], which, based on in vitro models, confirmed the regulation of triglyceride and cholesterol metabolism by miR-21 through the inhibition of 3-hydroxy-3-methylglutaryl-CoA reductase (HMGCR) expression [58].

No correlation was found between miR-21 and potential markers of increased intestinal barrier permeability, such as LBP and DAO, which suggests that the role of miR-21 in MASLD development is multifactorial, and its association with intestinal permeability dysfunction in MASLD is less pronounced. A noticeable association with TNF-α suggests a link between upregulated miR-21 and the inflammatory process in MASLD patients.

MiR-29 has been postulated as a promising marker involved in the regulation of intestinal barrier function. Studies in patients with the diarrheal subtype of IBS have shown that miR-29a downregulates zonula occludens-1 (ZO-1) and claudin-1 (CLDN1) [17], while both miR-29a and miR-29b reduce the expression of CLDN1 and NF-κB repressing factor (NKRF) in intestinal tissue [59]. Another study demonstrated that increased circulating levels of miR-29a in the blood correlate with increased intestinal barrier permeability, as assessed by the lactulose–mannitol test in patients with irritable bowel syndrome. Furthermore, changes in intestinal membrane permeability in these patients are mediated by miR-29a through glutamine-dependent pathways [60].

In our study, we did not observe an upregulation of miR-29a in patients with MASLD compared to the healthy control group. Moreover, miR-29a levels did not correlate with any of the other potential markers of increased intestinal permeability that we analyzed. The observed slight negative correlation with serum DAO levels was not clinically significant, despite previously reported association between miR-29a and DAO activity in the intestinal mucosa of mouse models [17].

Chronic low-grade inflammation resulting from metabolic dysregulation and increased intestinal permeability is closely associated with the progression of MASLD and MASH. TNF-α, a proinflammatory cytokine released during the immune response, plays a key role in both of these conditions [61]. Persistent liver exposure to IL-6 can contribute to hepatic insulin resistance [62], while IL-6 trans-signaling and increased hepatic IL-6 expression have been linked to liver stiffness and fibrosis in fatty liver disease [63].

Moreover, in the gastrointestinal tract, IL-6 has been shown to regulate immune responses to environmental microorganisms, increase intestinal epithelial tight-junction permeability by enhancing claudin-2 expression, and promote intestinal epithelial proliferation during mucosal repair in both physiological and pathological conditions [64].

In contrast to aforementioned studies, we did not observe significant differences in serum TNF-α or IL-6 levels between the study and control groups. Nevertheless, the associations of miR-21 and miR-122 with TNF-α suggest that their upregulation may reflect inflammatory activity in MASLD. Consistently, miR-122-5p has been implicated in promoting hepatic inflammation and oxidative stress through inhibition of FOXO3, thereby contributing to NAFLD progression [65].

There was no correlation found between IL-6 and any of the miRNAs, contrary to what has been reported in some previous studies [66].

This study has several limitations. The primary limitation is the small study subgroups, with only 15 MASLD patients diagnosed with liver fibrosis. It cannot be ruled out that in a larger cohort, the observed associations might be more pronounced. Therefore, evaluating these markers in a larger patient group would be appropriate.

Additionally, the study does not include follow-up data from patients, which could have further strengthened the findings.

Another important limitation is the absence of a distinct group of patients with MASH. Assessing the selected markers in this patient population could further validate their significance in the disease pathomechanism, including their role in intestinal barrier dysfunction in MASH development [25]. However, a definitive diagnosis requires liver biopsy for a rigorous assessment of MASH. Due to the invasive nature of liver biopsy, potential adverse events, and sampling variability, we opted to assess MASH-related fibrosis using transient elastography in this study.

Similarly, for MASLD, where liver biopsy still remains the gold standard for diagnosis, we chose to evaluate hepatic steatosis using CAP measured concurrently with liver stiffness assessment via TE using the FibroScan^®^ device. This non-invasive technique is a reliable, biopsy-validated tool for assessing MASLD progression. By evaluating a larger liver area, it reduces sampling error. CAP is recognized as an accurate tool for assessing liver steatosis, particularly for the early detection of mild stages [67]. Numerous studies are currently evaluating its effectiveness in determining the degree of liver steatosis, utilizing appropriate probes selected based on BMI and an automated probe selection tool [68].

As forementioned, some of selected biomarkers for this study are postulated to be markers of disrupted intestinal barrier. In current study we did not use standardized tests for intestinal permeability assessment, such as lactulose/mannitol ratio mainly due to project of the study, which was based on collected serum of previously examined patients. A thorough assessment of intestinal barrier function could provide a valuable extension to the present study by addressing this pathomechanistic pathway, especially since the utility of commonly used serum markers of intestinal permeability—such as intestinal fatty acid binding protein (I-FABP), zonulin, or LPS/LBP—remains a matter of debate [10,69].

## 4. Materials and Methods

### 4.1. Study Participants and Clinical Assessment

For this study, we recruited consecutive patients with the diagnosis of MASLD, previously described in our original article [70]. In brief, between March 2018 and February 2020, a total of 127 adults were consecutively invited to participate in this prospective study conducted at a single outpatient center in Szczecin, Poland. After obtaining written informed consent, physical examination and complete medical histories of all participants were collected along with questionnaires regarding present and past alcohol consumption. Demographic and anthropometric measurements were obtained, including BMI and waist and hip circumferences.

The diagnosis of MASLD was established based on multisociety Delphi consensus on new fatty liver disease nomenclature [2]. Patients with history of significant alcohol consumption (exceeding 30 g per day for men and 20 g per day for women at any point in their lives) or other known chronic liver diseases, including viral hepatitis (seropositivity for hepatitis B surface antigen or anti-hepatitis C virus antibodies), other recognized causes of chronic liver disease (infectious, iatrogenic, and inherited) were excluded. Additionally, patients with major systemic illnesses, including genetic, autoimmune, or acquired disorders and chronic kidney disease were also not considered for this study.

A control group of healthy volunteers (*n* = 57, 20 males, median age 54 years) were also included. These patients were screened negatively for liver steatosis in ultrasonography and CAP examination (see description below), and had no other clinical conditions that may influence the study results.

### 4.2. Liver Stiffness and CAP Measurements

Evaluation of liver stiffness and the grade of liver steatosis was performed using the FibroScan^®^ device (Echosens, Paris, France) by single investigator (KK). Liver steatosis was assessed by CAP parameter, using the cut-off values established by Karlas et al. [18]. CAP measurement is considered a reliable, quantitative tool that has been thoroughly validated against liver biopsy. It is also less prone to sampling size error due to larger part of liver tissue examined. CAP is regarded as an accurate tool for liver steatosis assessment, especially for the early detection of mild steatosis [67]. The following cut-off values were used for determining steatosis grades: low (S1) 234 dB/m, intermediate (S2) 269 dB/m, and high (S3) 301 dB/m [18]. Measurements were acquired with both M (3.5 MHz) and XL (2.5 MHz) probes, selected based on the skin-to-liver capsule distance (≤25 mm or >25 mm). The choice of the probe was guided by an integrated tool.

Fibrosis was assessed using TE with a FibroScan^®^ device. TE is a non-invasive, reliable alternative to liver biopsy tool for evaluating liver fibrosis, particularly effective in identifying advanced stages [68].

For additional steatosis and fibrosis assessment, HRI, FLI, as well as FIB-4 and APRI indexes were calculated from data obtained at the same appointment as ultrasound, elastography, and CAP examination. HRI defined by the echogenicity ratio of liver to right renal cortex serves as a screening parameter for hepatic steatosis, widely validated against liver biopsy [71]. FLI, which incorporates BMI, waist circumference, triglycerides, and GGT, has also been described as an accurate predictor of hepatic steatosis [72]. FIB-4 calculated from patient age, aminotransferases levels, and platelet count, is widely applied as a non-invasive tool for detecting advanced liver fibrosis in numerous liver diseases, including MASLD. Similarly, APRI, based on AST and platelet count, demonstrates high diagnostic accuracy in assessing liver fibrosis not only in patients with HBV or HCV infection but also in MASLD [73].

### 4.3. Biomarkers of Intestinal Permeability

For the analysis of selected biomarkers of intestinal permeability, we used fasting serum samples collected simultaneously with clinical and FibroScan^®^ examinations from patients and healthy controls. The samples were stored at a stable temperature of −80 °C until analysis, in accordance with standard protocols for long-term preservation of serum biomarkers to ensure optimal sample integrity. Blood was collected into serum-separating tubes, allowed to clot for 30 min at room temperature, and centrifuged at 1500× *g* for 10 min. Serum was aliquoted into 200 µL RNase-free tubes to avoid repeated freeze–thaw cycles. Before analyses, samples were thawed on ice, gently mixed, and centrifuged again at 10,000× *g* for 5 min to remove residual debris.

### 4.4. ELISA Analyses

LBP concentration was measured using a Human LBP ELISA kit (EH297RB, Invitrogen, Thermo Fisher Scientific, Waltham, MA, USA); serum DAO concentration was measured using Human DAO (Diamine Oxidase) ELISA kit (E-EL-H1241, Elabscience, Houston, TX, USA); serum TNF-α concentration was measured using Human TNFa (Tumor Necrosis Factor Alpha) ELISA kit (ELK1190, ELK Biotechnology, Sugar Land, TX, USA); and IL-6 serum concentrations was measured using Human IL6 (Interleukin 6) ELISA kit (ELK1156, ELK Biotechnology, Sugar Land, TX, USA), according to the manufacturer’s protocols. All analyses were performed in duplicate. Serum was diluted 1:100 for LBP, whereas DAO, TNF-α, and IL-6 were assayed undiluted unless optical densities exceeded the standard curve range. A total of 100 µL of standards, blanks, and serum samples were added to each well and incubated for 1 h at 37 °C. After 3–5 washing steps, 100 µL of detection antibody was added for 1 h at 37 °C, followed by HRP-conjugate incubation for 30 min. After adding TMB substrate for 10–15 min in the dark, the reaction was stopped and absorbance was read at 450 nm. Standard curves were generated using a four-parameter logistic regression, and sample concentrations were calculated automatically using GraphPad Prism version 7.0 software (GraphPad Software, San Diego, CA, USA). Intra-assay CV < 10% was accepted.

### 4.5. miRNA Expression Analysis

Total RNA was isolated from patients’ serum using miRNeasy Serum/Plasma Advanced kit (Qiagen, Hilden, Germany) following the manufacturer’s protocol. Complementary DNA (cDNA) was synthesized from the extracted RNA using the TaqMan Advanced miRNA cDNA Synthesis Kit (Applied Biosystems, Thermo Fisher Scientific, Waltham, MA, USA), according to the manufacturer’s protocol. The relative expression levels of selected microRNAs were determined by quantitative real-time PCR (qRT-PCR). miR-21 (477975_mir), miR-29a (478002_mir), miR-122 (477855_mir), and the endogenous control miR-16 (477860_mir) were quantified using Taq Man Fast Advanced Master Mix (Applied Biosystems, Waltham, MA, USA). Reactions were carried out in 96-well plates on a QuantStudio 5 Real-Time PCR System (Applied Biosystems, Waltham, MA, USA), with each sample analyzed in technical triplicates. Threshold cycle (Ct) values were automatically determined by the system software, and relative expression levels of target microRNAs were calculated using the 2^−ΔΔCt^ method with miR-16 serving as the reference control. No exogenous spike-in controls were used; all analyses were normalized exclusively to the endogenous control miR-16. qPCR reactions were performed in a total volume of 10 µL, consisting of 5 µL TaqMan Fast Advanced Master Mix, 0.5 µL TaqMan probe, 2.5 µL diluted cDNA, and 2.0 µL nuclease-free water. The cycling conditions were as follows: initial enzyme activation at 95 °C for 20 s, followed by 40 cycles of denaturation at 95 °C for 3 s, and annealing/extension at 60 °C for 30 s.

### 4.6. Ethics

The study protocol for this observational, non-interventional research was reviewed and approved by the Ethics Committee of the Pomeranian Medical University in Szczecin, Poland (approval no. KB-0012/08/18; issued on 5 February 2018). The investigation was conducted in accordance with the ethical standards set forth in the 1975 Declaration of Helsinki and its 2013 revision. Written informed consent was obtained from all participants for participation and for the publication of the study results.

### 4.7. Statistical Analysis

#### 4.7.1. Descriptive Statistics

Descriptive statistical methods were used to analyze the study group and serum concentrations of analyzed factors. Categorical data were described using the number of observations and absolute frequencies. Continuous variables were presented as median and range as they showed non-normal distributions.

#### 4.7.2. Comparative Analyses

Differences between subgroups were assessed using the Mann–Whitney U test or the Kruskal–Wallis rank-sum test, as appropriate. Comparisons of prevalence between groups were conducted using a two-tailed Fisher’s exact test.

#### 4.7.3. Correlation and Regression

Correlations were evaluated with Spearman’s rank correlation coefficient. To determine independent predictors of liver steatosis severity and significant fibrosis, a multivariable logistic regression analysis with stepwise backward elimination was performed. Due to the exploratory nature of the present study, the statistical tests were used primarily for hypothesis-generating purposes. Therefore, no additional multiple comparison correction procedures (e.g., Bonferroni or Benjamini–Hochberg FDR) were applied. Variables showing significant associations in the univariable analysis (*p* < 0.05) were included in the model, encompassing both clinical and laboratory parameters.

#### 4.7.4. Software

All statistical analyses and graphical outputs were generated using Stata version 18.0 (StataCorp LLC, College Station, TX, USA; 2023). A value of *p* < 0.05 was considered statistically significant.

## 5. Conclusions

In this study, we identified elevated serum miR-122, as well as downregulated miR-21, as independent predictors of MASLD. Increased serum diamine oxidase (DAO) emerged as an independent predictor of steatosis severity.

These parameters may serve as a foundation for the development of novel non-invasive approaches to the assessment of patients with MASLD, for example, by contributing to new diagnostic indices.

## Figures and Tables

**Figure 1 ijms-26-11351-f001:**
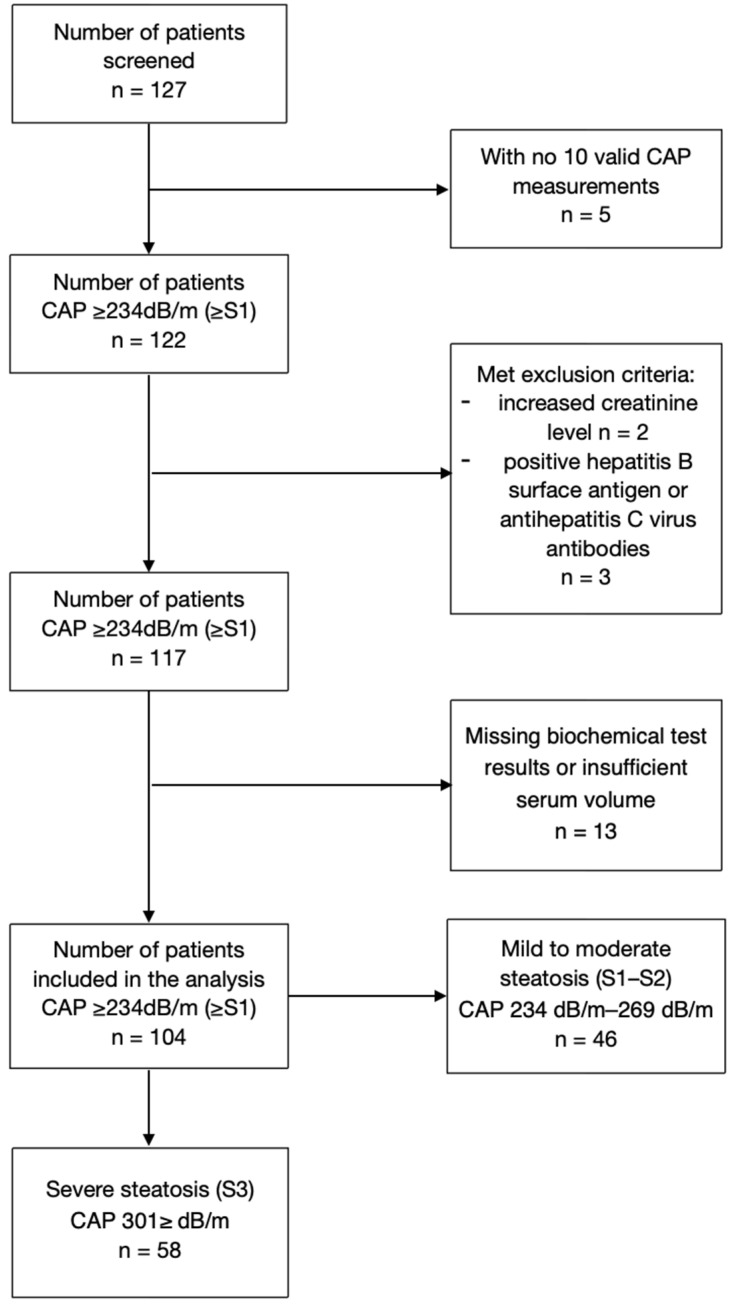
Study population.

**Figure 2 ijms-26-11351-f002:**
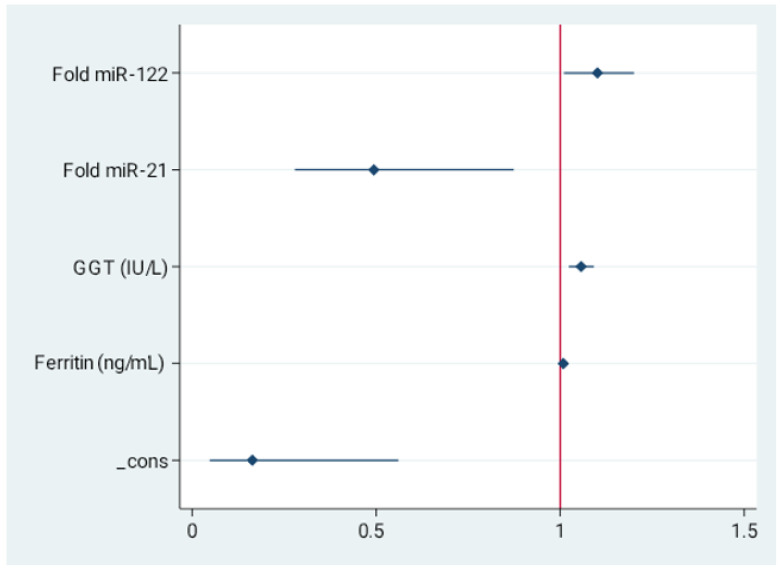
Independent indicators of the presence of liver steatosis. Forest plot for multivariable stepwise logistic regression. Values presented as odds ratio (OR) and 95% confidence interval (CI). Final model revealed increased miR-122, GGT, and ferritin as well as decreased miR-21 as independent predictors of MASLD.

**Figure 3 ijms-26-11351-f003:**
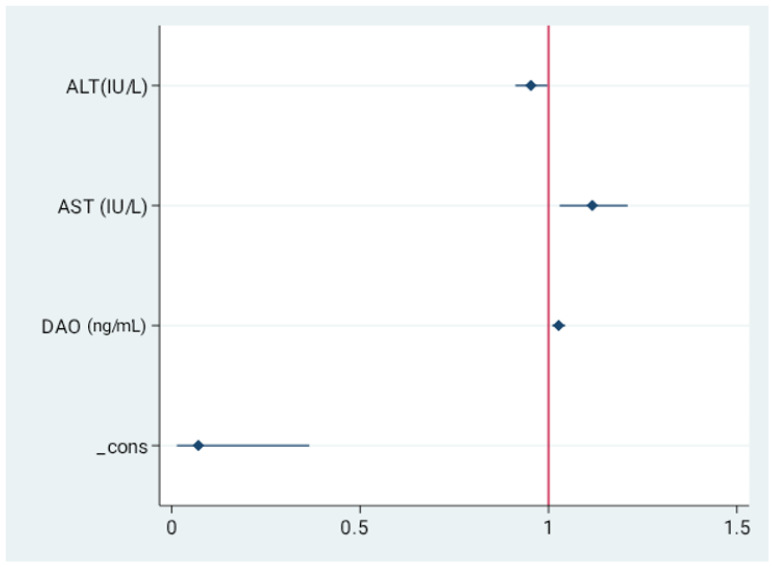
Independent predictors for the presence of severe liver steatosis. Values presented as odds ratio (OR) and 95% confidence interval (CI). Final model revealed serum concentrations of AST, ALT, and DAO as independent predictors of severe steatosis.

**Table 1 ijms-26-11351-t001:** Descriptive statistics and comparison of study groups. Data presented as median (range). Two-sample Wilcoxon rank-sum (Mann–Whitney) test and Fisher’s exact test. *p* values < 0.05 taken as significant. The abbreviation ‘n/a’ denotes entries for which the variable is not available.

Variable	Study Group (*n* = 104)	Control Group (*n* = 57)	*p* Value
Male gender	42 (40.4%)	20 (35.1%)	0.61
Age (years)	55 (20–83)	54 (28–81)	0.67
BMI (kg/m^2^)	30.3 (21.8–47.3)	24.1 (18.7–33.7)	<0.0001
Laboratory tests			
ALT (IU/L)	21 (6–205)	14 (9–40)	0.0003
AST (IU/L)	24 (12–114)	18 (12–47)	0.0004
GGT (IU/L)	36 (9–235)	17 (4–92)	<0.0001
Total bilirubin (mg/dL)	0.4 (0.1–2.7)	0.3 (0.1–1.6)	0.21
Albumin (g/L)	49 (37–57)	48 (41–71)	0.60
Glucose (mg/dL)	98 (80–283)	92 (73–118)	0.0004
Insulin (U/mL)	13.9 (2.5–224)	7.5 (1.9–52.7)	<0.0001
Ferritin (μg/L)	185 (7.1–1254)	77.2 (3.3–282)	<0.0001
Total cholesterol (mg/dL)	202 (117–290)	207 (139–322)	0.09
LDL (mg/dL)	132 (56–222)	128 (66–249)	0.86
Triglycerides (mg/dL)	146 (49–521)	98 (49–205)	<0.0001
Noninvasive markers of steatosis			
CAP	310 (235–400)	213 (144–233)	<0.0001
HRI	1.93 (1.1–3.6)	1.27 (0.97–2.01)	<0.0001
FLI	79.5 (11.5–99.7)	27.9 (2.5–87.8)	<0.0001
Noninvasive markers of fibrosis			
TE (kPa)	5.0 (2.9–19.7)	4.4 (2.1–6.3)	0.0001
FIB-4	1.2 (0.3–5.0)	n/a	
APRI	0.002 (0.001–0.02)	n/a	

**Table 2 ijms-26-11351-t002:** Comparison of analyzed parameters in study groups. Data presented as median (range). Two-sample Wilcoxon rank-sum (Mann–Whitney) test and Fisher’s exact test. *p* values < 0.05 taken as significant.

Tested Biomarkers	Study Group (*n* = 104)	Control Group (*n* = 57)	*p* Value
LBP (μg/mL)	28.8 (0.08–120.4)	19.5 (6.1–51.0)	0.002
TNFa (pg/mL)	7.7 (0.03–76.6)	7.6 (0.2–29.6)	0.99
IL-6 (pg/mL)	3.2 (0.07–29.4)	2.7 (0.1–81.9)	0.26
DAO (ng/mL)	41.6 (0.7–209.3)	34.4 (1.2–216.6)	0.15
Fold miR-21	0.9 (0.03–17.3)	1.1 (0.06–7.97)	0.07
Fold miR-122	5.3 (0.09–1721.5)	1.0 (0.07–48.17)	<0.0001
Fold miR-29a	1.6 (0.05–403.0)	0.9 (0.01–50.4)	0.18

**Table 3 ijms-26-11351-t003:** Correlations between analyzed parameters and laboratory and clinical variables in study group. Spearman’s correlation. *p* < 0.05 is taken significant.

Variable	DAO		TNFa		IL-6		LBP		miR-122		miR-21		miR-29a	
	Rho	*p* Value	Rho	*p* Value	Rho	*p* Value	Rho	*p* Value	Rho	*p* Value	Rho	*p* Value	Rho	*p* Value
Age	−0.08	0.42	−0.12	0.21	−0.07	0.50	0.07	0.49	−0.17	0.08	0.01	0.89	0.09	0.37
BMI	0.05	0.60	0.01	0.88	0.04	0.70	0.10	0.31	−0.10	0.34	0.004	0.96	0.06	0.51
Laboratory tests														
ALT	0.01	0.88	0.09	0.36	−0.18	0.07	0.03	0.74	0.40	<0.0001	0.07	0.50	−0.10	0.30
AST	−0.18	0.07	0.16	0.11	−0.15	0.12	−0.006	0.95	0.50	<0.0001	0.20	0.04	0.07	0.49
GGT	−0.03	0.78	0.05	0.58	−0.21	0.04	0.14	0.15	0.27	0.005	0.09	0.34	−0.03	0.78
BIL-T	−0.07	0.44	0.13	0.17	−0.04	0.71	−0.25	0.01	0.21	0.03	0.13	0.20	−0.01	0.92
Albumin	0.05	0.59	0.15	0.13	−0.03	0.79	−0.17	0.09	0.18	0.07	0.09	0.33	−0.04	0.67
INR	0.15	0.25	−0.06	0.63	0.33	0.009	−0.07	0.62	−0.11	0.40	−0.11	0.41	−0.04	0.77
Glucose	−0.02	0.86	−0.19	0.08	−0.08	0.45	0.14	0.21	−0.17	0.11	−0.19	0.08	−0.20	0.06
Insulin	0.02	0.83	0.03	0.72	−0.05	0.62	0.14	0.17	0.13	0.18	−0.04	0.73	−0.13	0.19
Ferritin	0.04	0.69	0.10	0.32	−0.03	0.79	−0.06	0.54	0.20	0.04	0.05	0.56	−0.01	0.91
Chol-T	−0.07	0.51	−0.07	0.46	−0.18	0.07	0.15	0.14	0.07	0.50	0.10	0.29	0.03	0.75
LDL	−0.01	0.90	−0.03	0.77	−0.21	0.04	0.19	0.05	0.16	0.10	0.17	0.07	0.08	0.44
TG	0.006	0.95	0.07	0.49	0.04	0.72	0.16	0.11	0.01	0.89	−0.07	0.50	−0.11	0.26
Non-invasive markers of steatosis														
CAP	0.22	0.02	0.04	0.67	−0.12	0.22	0.23	0.02	0.17	0.08	0.03	0.77	−0.12	0.22
HRI	0.03	0.74	0.07	0.49	0.01	0.91	−0.06	0.55	0.22	0.02	0.15	0.12	−0.08	0.45
FLI	0.03	0.76	0.14	0.17	−0.12	0.26	0.16	0.13	0.28	0.005	0.09	0.37	0.08	0.43
Non-invasive markers of fibrosis														
TE	−0.12	0.23	0.13	0.19	−0.16	0.10	0.11	0.26	0.22	0.03	0.15	0.14	0.05	0.59
APRI	−0.26	0.02	0.01	0.89	−0.10	0.38	0.07	0.55	0.41	0.0001	0.09	0.40	−0.05	0.67
FIB-4	−0.37	0.0007	−0.12	0.26	−0.04	0.71	0.07	0.55	0.16	0.16	0.06	0.61	0.03	0.78

**Table 4 ijms-26-11351-t004:** Variables vs. steatosis and fibrosis severity. Two-sample Wilcoxon rank-sum (Mann–Whitney) test.

Variable	Advanced Steatosis (CAP Karlas 2014 [18])			Significant Fibrosis (TE)		
	S1–S2 (*n* = 46)	S3 (*n* = 58)	*p* Value	F0–F1 (*n* = 87)	F2–F4 (*n* = 15)	*p* Value
ALT (IU/L)	18 (6–205)	21 (10–103)	0.02	20 (6–205)	29 (14–103)	0.02
AST (IU/L)	20 (12–114)	25 (14–74)	0.0009	22 (12–114)	32 (19–74)	0.0005
GGT (IU/L)	33 (9–139)	39.5 (15–235)	0.04	35 (9–235)	44 (15–139)	0.11
ALP (IU/L)	74 (32–162)	74 (45–167)	0.79	73 (32–162)	84 (49–167)	0.11
Bilirubin (mg/dL)	0.4 (0.1–1.4)	0.4 (0.2–2.7)	0.55	0.4 (0.1–2.7)	0.4 (0.2–1.3)	0.60
Glucose (mg/dL)	98 (80–127)	99 (80–283)	0.28	98 (80–283)	99 (89–178)	0.69
Insulin (U/mL)	18 (2–55)	16 (6–224)	0.0001	13 (2–57)	27 (8–224)	0.0006
Ferritin (μg/L)	134 (7–487)	191 (16–1254)	0.08	170 (7–993)	199 (20–487)	0.58
Chol-T (mg/dL)	196 (120–287)	205 (117–290)	0.62	202 (117–290)	208 (119–269)	0.83
LDL (mg/dL)	127 (65–199)	133 (56–222)	0.90	133 (56–222)	131 (58–188)	0.74
TG (mg/dL)	131 (49–502)	158 (57–521)	0.05	138 (49–521)	161 (100–245)	0.20
LBP (μg/mL)	22.9 (6.4–78.5)	33.2 (0.08–120.4)	0.04	28.8 (0.08–120.4)	27.0 (11.6–107.8)	0.70
TNFa (pg/mL)	6.8 (0.03–76.1)	7.8 (0.03–76.6)	0.49	7.7 (0.03–76.6)	9.5 (0.5–27.5)	0.18
IL-6 (pg/mL)	3.7 (0.1–29.4)	2.8 (0.07–24.3)	0.26	3.2 (0.1–29.4)	3.0 (0.07–17.5)	0.47
DAO (ng/mL)	38.9 (0.07–113.6)	49.0 (12.8–209.3)	0.04	44.2 (0.7–209.3)	28.9 (12.8–71.4)	0.04
Fold miR-21	0.85 (0.05–16.5)	0.97 (0.03–17.2)	0.47	0.9 (0.03–17.3)	0.9 (0.04–4.2)	0.45
Fold miR-122	4.6 (0.2–214.1)	6.7 (0.09–1721.5)	0.12	4.8 (0.09–1721.5)	7.6 (0.4–68.3)	0.05
Fold miR-29a	1.7 (0.07–66.4)	1.4 (0.05–403.0)	0.78	1.9 (0.05–403.0)	1.0 (0.05–114.7)	0.76

## Data Availability

The original contributions presented in this study are included in the article/Appendix A. Further inquiries can be directed to the corresponding author.

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
