# Peer review of "MicroRNA and Protein Biomarkers of Intestinal Permeability in the Assessment of Metabolic Dysfunction-Associated Steatotic Liver Disease (MASLD)"

_ijms, 2025, doi:10.3390/ijms262311351_

Round 1

Reviewer 1 Report

Comments and Suggestions for Authors

1.It is unclear why only a limited set of predefined protein markers and miRNAs were selected for analysis instead of employing a more comprehensive and unbiased approach such as proteomics or miRNA sequencing.

2. LBP was only shown to be associated with CAP, but no correlation was observed with BMI, insulin, or other metabolic indicators. Although this was mentioned in the Discussion section, no in-depth analysis or discussion was provided.

3. DAO, as a marker of intestinal barrier disruption, showed opposite associations in liver steatosis and liver fibrosis — in particular, it was decreased in patients with fibrosis. This finding was not thoroughly discussed in the Discussion section.

4.Given the large number of multiple hypothesis tests performed, controlling for the inflation of Type I error is necessary. I recommend applying multiple comparison correction methods, such as Benjamini‑Hochberg FDR or Bonferroni adjustment, to ensure the robustness of the reported associations.

Comments on the Quality of English Language

The quality of English in the manuscript is generally acceptable, but there are occasional grammatical errors and awkward phrasings that require revision. A thorough language polishing by a native or professional editor is recommended to improve clarity and readability.

Author Response

Thank you for your comments concerning article “MicroRNA and protein biomarkers of intestinal permeability in the assessment of metabolic dysfunction-associated steatotic liver disease (MASLD)”. Below, we include our responses to the comments and proposed revisions.

Comment 1: It is unclear why only a limited set of predefined protein markers and miRNAs were selected for analysis instead of employing a more comprehensive and unbiased approach such as proteomics or miRNA sequencing.

Response 1: Thank you for your insightful comment; we fully agree with your point. In this study, we selected only a limited set of protein markers and microRNAs based on previous research investigating miRNA expression and protein markers in patients with a disrupted intestinal barrier. Studies examining a broader spectrum of markers in MASLD patients are planned for future work.

Comment 2: LBP was only shown to be associated with CAP, but no correlation was observed with BMI, insulin, or other metabolic indicators. Although this was mentioned in the Discussion section, no in-depth analysis or discussion was provided.

Response 2: Thank you for this comment. We have provided additional paragraph concerning this matter in sections 284-293 (p.10).

Comment 3: DAO, as a marker of intestinal barrier disruption, showed opposite associations in liver steatosis and liver fibrosis — in particular, it was decreased in patients with fibrosis. This finding was not thoroughly discussed in the Discussion section.

Response 3: Thank you for this guidance. We have provided additional paragraph concerning this matter in sections 315-321 (p.11).

Comment 4: Given the large number of multiple hypothesis tests performed, controlling for the inflation of Type I error is necessary. I recommend applying multiple comparison correction methods, such as Benjamini‑Hochberg FDR or Bonferroni adjustment, to ensure the robustness of the reported associations.

Response 4: Thank you for pointing out the important issue of potential Type I error inflation due to the large number of statistical tests performed. In the present study, however, we did not apply additional multiple comparison corrections. This decision was based on the exploratory nature of our work, which aimed to identify potential associations rather than to draw definitive confirmatory conclusions. Moreover, the multivariable regression models used in our analysis inherently account for the simultaneous influence of multiple predictors, which reduces the risk of drawing misleading conclusions solely from univariable comparisons. For this reason, we did not introduce separate correction procedures. We have clarified methodological approach in the revised “Statistical analysis” section to ensure transparency in the analytical strategy applied (page 16).

Reviewer 2 Report

Comments and Suggestions for Authors

The study examined the correlation between the severity of MASLD in a well-characterized patient group and serum markers of intestinal permeability (lipopolysaccharide binding protein, diamine oxidase, interleukin 6, and tumor necrosis factor). The authors are advised to take the following points into consideration.

  1. Previous studies have addressed non-invasive biomarkers in MASLD patients, and the authors should clarify the novelty and contribution of their current study.
  2. Not all markers used are indicators of intestinal permeability. TNF-α and IL-6 are indicators of systemic inflammation, while LBP and DAO are indicators of intestinal permeability. This must be taken into account, and the entire manuscript must focus on the specific intestinal permeability indicators.
  3. Please mention in the introduction the markers of intestinal permeability and refer to the studies that considered micro-RNA as markers of intestinal permeability.
  4. The introduction section needs to explain the relationship between intestinal permeability and MASLD, and how intestinal permeability affects the MASLD.
  5. The introduction section also needs to mention the non-invasive markers that were addressed in previous studies and the difference between them and the markers chosen in the current study.
  6. It is preferable to move the objective of the study (lines 67-74) to the end of the introduction section.
  7. The tables need improvement, especially Table 3.
  8. In the Materials and Methods section, please explain how to perform all the analyses mentioned in the study.
  9. The conclusion must primarily reflect the main point of the manuscript, which is the changes in intestinal permeability markers in cases of MASLD, and therefore their potential use as non-invasive indicators. Ferritin and GGT are not intestinal permeability markers.

Author Response

Thank you for your comments concerning article “MicroRNA and protein biomarkers of intestinal permeability in the assessment of metabolic dysfunction-associated steatotic liver disease (MASLD)”. Below, we include our responses to the comments and proposed revisions.

Comment 1: Previous studies have addressed non-invasive biomarkers in MASLD patients, and the authors should clarify the novelty and contribution of their current study.

Response 1: Thank you for this important suggestion. The clarification of the novelty and contribution of this study has been added (page 3).

Comment 2: Not all markers used are indicators of intestinal permeability. TNF-α and IL-6 are indicators of systemic inflammation, while LBP and DAO are indicators of intestinal permeability. This must be taken into account, and the entire manuscript must focus on the specific intestinal permeability indicators.

Response 2: Thank you for this correction. It has been emphasized that TNF-α and IL-6 are inflammatory markers, not indicators of intestinal permeability (page 3).

Comment 3: Please mention in the introduction the markers of intestinal permeability and refer to the studies that considered micro-RNA as markers of intestinal permeability.

Response 3: Thank you for this comment. Proposed information was added in sections 72-89 and 105-110 (p.2-3).

Comment 4: The introduction section needs to explain the relationship between intestinal permeability and MASLD, and how intestinal permeability affects the MASLD.

Response 4: Thank you for pointing this out. The explanation has been provided in sections 60-65 (p.2).

Comment 5: The introduction section also needs to mention the non-invasive markers that were addressed in previous studies and the difference between them and the markers chosen in the current study.

Response 5: Thank you for this suggestion. The additional information was provided in section 72-89 (p.2).

Comment 6: It is preferable to move the objective of the study (lines 67-74) to the end of the introduction section.

Response 6: Thank you for this guidance. The section has been moved at the end of the introduction section (p.3).

Comment 7: The tables need improvement, especially Table 3.

Response 7: Thank you for this correction. The tables have been corrected. (pages 4-7)

Comment 8: In the Materials and Methods section, please explain how to perform all the analyses mentioned in the study.

Response 8: Thank you for the comment. Detailed methodology of performed analyses has been added. (p.14-15)

Comment 9: The conclusion must primarily reflect the main point of the manuscript, which is the changes in intestinal permeability markers in cases of MASLD, and therefore their potential use as non-invasive indicators. Ferritin and GGT are not intestinal permeability markers.

Response 9: We appreciate this comment. The conclusion section has been revised. (p.16)

Round 2

Reviewer 2 Report

Comments and Suggestions for Authors

The manuscript has been revised and modified according to the comments presented previously.